# Demystifying Limited Adversarial Transferability in Automatic Speech Recognition Systems

**Hadi Abdullah, Aditya Karlekar, Vincent Bindschaedler, & Patrick Traynor**
University of Florida
{hadi10102,akarlekar,vbindschaedler,traynor}@ufl.edu

## Abstract

The *targeted transferability* of adversarial samples enables attackers to exploit black-box models in the real world. Optimization attacks are the most popular means of producing such transferable samples. This is because these samples have high levels of transferability in some domains. However, recent research has shown that samples from these attacks do not transfer when applied to Automatic Speech Recognition systems (ASRs). In this paper, we study this phenomenon, perform exhaustive experiments, and identify the factors that are preventing transferability in ASRs. To do so, we perform an ablation study on each stage of the ASR pipeline. We discover and quantify six factors (i.e., input type, MFCC, RNN, output type, and vocabulary and sequence sizes) that impact the targeted transferability of optimization attacks against ASRs. Our findings can be leveraged to design ASRs that are more robust to other transferable attack types (e.g., signal processing attacks), or to modify architectures in other domains to reduce their vulnerability to targeted transferability.

## 1 Introduction

It is hard to understate the pervasiveness of ASRs in security-critical systems. These include banking (Amazon, 2021), surveillance (Froomkin, 2015), online retail (Vigderman, 2021), and home security (Bharadwaj, 2019). However, ASRs are vulnerable to specially crafted adversarial samples, that force them to produce malicious outputs. In the research community, the most popular class of algorithms for this purpose are the optimization (or gradient-based) attacks (Abdullah et al., 2021b). This is largely because these attacks exhibit *targeted transferability* in some domains. Specifically, samples crafted for a local model (surrogate) can force a different black-box model (target) to produce the attacker chosen output. This is the case even if the surrogate and target use different architectures, training data, hyper-parameters, etc. Therefore, the transferability property has enabled attackers to exploit security-critical applications which include facial recognition systems (Shan et al., 2020), image APIs (Brown et al., 2017; Liu et al., 2016), authentication systems (Chen et al., 2019), and malware detectors (Hu & Tan, 2017; Kreuk et al., 2018b; Grosse et al., 2017). In short, target transferability makes optimization attacks effective against several real-world systems.

However, recent work has uncovered that these attacks do not exhibit target transferability between ASRs, even when the ASRs are trained on *identical* setups (i.e., same hyper-parameters, architecture, random seed, training data) (Abdullah et al., 2021b). Therefore, these attacks can not be used against black-box ASRs, casting doubt on the usefulness of the entire class of optimization attacks in the audio domain. However, the reasons for this failure is not understood.

In this work, we study this phenomenon to uncover the factors that prevent the transferability of optimization attacks between ASRs. To ensure we can uncover each one of these factors, we perform an exhaustive ablation study on the entire ASR pipeline and observe the impact of the different components on the transferability rate. We test thousands of adversarial samples across multiple models and characterize six factors that impact the transferability rate. In doing so, we make the following contributions:

1. We identify six previously unknown factors that impact target transferability. These include the input type, Mel Frequency Cepstral Coefficient (MFCC), the Recurrent Neural Network (RNN), output type, and the vocabulary and sequence sizes. These factors explain the near 0% transferability rate seen in prior works.

2. We highlight the relationship between accuracy and adversarial robustness in ASRs. The five factors that improve ASR robustness *also* improve accuracy.

3. Our findings explain why one of the most popular classes of attacks, across the adversarial machine learning space, fails in the audio domain. These can be leveraged to strengthen models from other domains (e.g., images), that have traditionally been vulnerable to optimization attacks.

We begin our study by listing all the factors from the existing literature that are known to hinder transferability (Section 2.1). Even when controlling for these factors, transferability rates in ASRs still do not achieve the near 100% observed in the image models (Section 4.1). This suggests the existence of additional factors limiting transferability. We list the additional potential factors (Section 2.3), describe our design choices (Section 3), and through a series of ablation experiments, we systematically quantify the impact of each factor on transferability (Section 4). Based on our findings, we discuss several takeaways (Section 5), present related work (Section 6), and summarize our findings (Section 7).

## 2 FACTORS

### 2.1 KNOWN FACTORS FROM EXISTING LITERATURE

Before delving into ASRs, it is first important to review the existing literature on transferability. While doing so, we identified 11 factors that are already known to limit transferability of optimization attacks:

1. Smoothness of gradients (Demontis et al., 2019; Zhou et al., 2018; Wu et al., 2018)
2. Attack type (Kurakin et al., 2016a; Dong et al., 2018; Liu et al., 2016)
3. Number of attack iterations (Dong et al., 2018)
4. Number of output labels (Liu et al., 2016)
5. Spectral makeup of the perturbations (Sharma et al., 2019; Guo et al., 2018)
6. Model architecture (Wu et al., 2018)
7. Model accuracy (Wu et al., 2018)
8. Model complexity (Demontis et al., 2019; Wu et al., 2018; Wu & Zhu, 2020)
9. Model agreement (Tramèr et al., 2017)
10. Confidence of the adversarial sample (Abdullah et al., 2021b)
11. Asymmetry (Wu et al., 2018; Wu & Zhu, 2020)

Unfortunately, outside of a single example (Abdullah et al., 2021b), the primary focus of most of these works has been image classification models. As a consequence, even when controlling for these factors, we observed that transferability between ASRs remains low (Section 4.1). This suggests the existence of unknown factors limiting transferability.

### 2.2 ASR PIPELINE

To identify these unknown factors, we first provide a brief overview of the different components that make up the ASR pipeline (Figure 1). To that end, we consider the most commonly attacked ASR pipeline in the research community (Abdullah et al., 2021b).

The first stage of an ASR splits the input audio into overlapping frames (Figure 1(i)). Next, a signal processing algorithm, (e.g., the MFCC (Lin & Abdulla, 2015)) extracts a feature vector from each of the overlapping frames (Figure 1(ii)). Next, the neural network (Figure 1 (iii)) assigns a single character label to each feature vector, resulting in a character list (e.g., "hheellllo bbbboooob") (Figure 1(iv)). These are then aggregated into a single word (e.g., "hello"), which is then combined with other words into a final sequence (e.g. "hello bob") (Figure 1(v)).

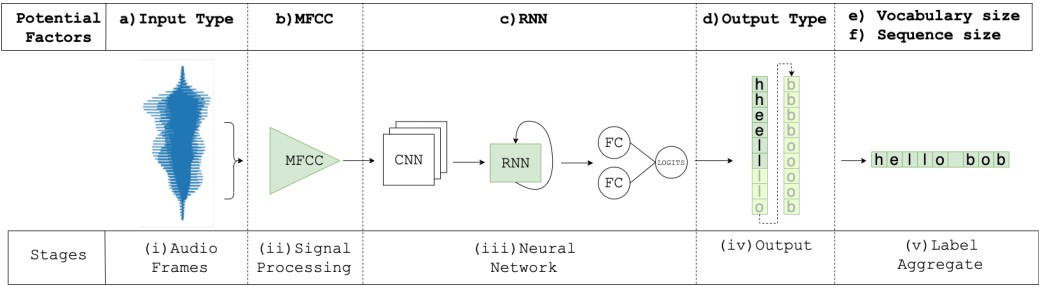

Figure 1: The typical ASR pipeline used in the adversarial research community. We can see the potential factors (a-f) and their locations at different stages (i-v) of the pipeline. We will study each of the impact of each potential factor on transferability.

## 2.3 ADDITIONAL POTENTIAL FACTORS

Having described the ASR pipeline, we can now list the components that might *potentially* impact transferability. We will later experimentally investigate their impact during ablation study. These potential factors include:

### 2.3.1 INPUT TYPE:

Given previous findings (Naseer et al., 2019), we hypothesize that the model's input type can impact transferability. For example, audio inputs are time-varying and one-dimensional. They do not contain topographical structures. In contrast, images are two-dimensional, topographical inputs with neighborhoods of pixels representing a single object (e.g., pixels representing a dog's snout). As a result, the input type will determine the model's decision boundaries, which can impact transferability.

### 2.3.2 MFCC:

We hypothesize that the MFCC algorithm could be a factor limiting transferability. This algorithm uses hand-crafted filters to regularize the feature vector and consequently, remove the high spectral noise (Mannell, 1994). Since regularization of the decision boundary or gradients has been shown to impact transferability (Demontis et al., 2019; Zhou et al., 2018; Wu et al., 2018; Wu & Zhu, 2020), we hypothesize that regularization of the feature vector using the MFCC plays a role as well.

### 2.3.3 RNN:

The network architecture can impact transferability as well. For example, the dropout layer, which is often included in model architectures for performance purposes, also inadvertently improves the transferability of adversarial samples (Demontis et al., 2019).

We hypothesize that the RNNis a factor limiting transferability in ASRs. This is because there are several RNN-based applications where authors have failed to demonstrate targeted transferability (Abdullah et al., 2021b).While these papers show transferability is hard in their respective domains, they do not contain definitive experiments studying why this is the case.

### 2.3.4 OUTPUT TYPE:

Since we know that output type can impact the transferability rate (Wei et al., 2018), we hypothesize that a sequential output type (instead of a single output label) can influence it as well. This type of output is commonly found in ASRs. For an adversarial sample to transfer, the victim ASR needs to assign the attacker chosen character to *each* frame of the adversarial audio. This might increase the chances of a mistake, which for even a few frames, could lead to a semantically wrong transcript (e.g., "cccaaatttt" vs "hhhaaatttt").

### 2.3.5 VOCABULARY AND SEQUENCE SIZES:

The last factor we consider is the output complexity or the total number of output labels. Transferability becomes harder as the number of output labels increases (Demontis et al., 2019; Wu et al., 2018; Wu & Zhu, 2020; Kurakin et al., 2016b). However, the output labels are not fixed in ASRs, as they are in the case of images (e.g., 10 output labels for the MNIST dataset). This is because ASRs are designed to output variable size sequences to account for the variation in the input audio length. As a result, instead of considering total output labels, we use two metrics to capture output complexity: vocabulary and sequence sizes (Figure 1(e) and (f)). The vocabulary size is the number of unique words in the training data that the ASR has learned to recognize. Similarly, the sequence size is the average number of words in each audio sample in the dataset. Both of these variables work in tandem to account for the ASR's output complexity.

## 3 STUDY DESIGN

Having outlined the six potential factors, we can now design an ablation study to measure their impact on transferability. Designing this study is non-trivial since we need to account for the 11 known factors from existing literature (Section 2.1) that limit transferability. If ignored, these factors alone can eliminate the transferability rate, hiding the effects of the aforementioned potential factors. We control for each of these 11 in our design.

Initially, we ran our ablation study on DeepSpeech, a real-world ASR which is commonly used in the adversarial research community. However, our experiments consistently yielded 0% transferability no matter what we did: removing or changing any of the ASR components did not change the transferability rate (Appendix A.2.). Upon further investigation, we realized this was primarily due to the large complexity of the model (a few million weights) and training data, which forced the transferability rate to remain unchanged 0%.

Overcoming this complexity was one of the major challenges of our work. We had to carefully design our experiments to remove the impact of complexity that comes with real-world ASRs, while simultaneously exposing the hidden factors limiting transferability. As a consequence, we had to run our study on *simple* yet realistic ASR designs to uncover the factors impacting transferability.

**Dataset:** Since transferability becomes harder with the total number of output labels (Demontis et al., 2019; Wu et al., 2018; Wu & Zhu, 2020; Kurakin et al., 2016b), we use the small Google Speech Commands dataset (Warden, 2018). This consists of clean, short audio files, each containing one of 30 unique labels. Each audio file is one second long (or a vector of size ∼16,000), either containing a number or a word. For the control experiment, we only choose a subset of the labels, specifically the numbers ONE to NINE (a total of nine labels).

**Model Architecture:** With this data, we can now train a simple number recognition ASR. We use the same architecture (Figure 1) typically found in existing adversarial research papers (Abdullah et al., 2021b)The model outputs one of nine labels for a single audio.

Additionally, we account for the known factors that hinder transferability. To do so, we train a small model of approximately 250,000 trainable parameters and do not include regularization (e.g., dropout) to limit the effects of complexity and regularization. To reduce model complexity further, instead of using complex RNN cell types (e.g., LSTMs (Gers et al., 1999) and GRUs (Chung et al., 2014)), we use the vanilla RNN cell (sim, 2021) (referred to as RNN in the remainder of the paper). We train five instances of the ASR on the exact *same* setup and hyper-parameters (architecture, random seed, epochs, batch size, training data slice, etc). Each of these is trained to the standard real-world ASR accuracy.

**Attack Formulation:** For our experiment, we can choose from several existing audio domain optimization attacks. Since some of these attacks are architecture specific (Abdullah et al., 2020), they have the potential to bias our results. As a consequence, we formulate a *generic* optimization attack that captures the intuition of existing works. This helps extend our results to all existing audio domain optimization attacks.

To do so, we first outline the steps used by optimization attacks, which generally follow the same approach. First, a perturbation $\delta$ is produced by minimizing:

$$||\delta||_2^2 + l(x + \delta, t) \tag{1}$$

where $l$ is the loss function, $x$ is the original audio sample, and $t$ is the target label. Next, $\delta$ is clipped by magnitude factor $\alpha \in \mathbb{R}$ to control the quality of the adversarial audio:

$$\delta = \text{clip}(\delta, \alpha, -\alpha) \tag{2}$$

Lastly, the $\delta$ is added to the original audio $x$ and clipped to create a valid sample $x_{\text{adv}}$:

$$x_{\text{adv}} = \text{clip}(\text{x} + \delta, 1, -1) \tag{3}$$

The clip step in Equation 2 is used to control the quality of the adversarial audio and is often specific to the target model architecture. For example, attacks that clip spectral gradients (Qin et al., 2019) can not work against end-to-end ASRs that do not have a spectrum generation step (Abdullah et al., 2020). Therefore, in our generic attack, we remove Equation 2 making it model agnostic.

**Adversarial Audio Generation:** We use this generic optimization attack to create adversarial samples for each of the five ASRs. Following prior work (Abdullah et al., 2021b; Liu et al., 2016), we only perturb audio samples that all of the ASRs transcribed correctly. We attack every label in the dataset since some sample labels could be easier to perturb and some are easier to transfer (Carlini et al., 2019). For example, we perturb an audio sample containing the label ONE to produce each of the remaining labels TWO to NINE. Running the experiments in this exhaustive manner allows us to generalize our findings.

We run the attack for 500 iterations. We save the adversarial sample every 50 iterations since the number of attack iterations can impact transferability (Dong et al., 2018). We also ensure that every saved adversarial sample has confidence greater than 0.99 because lower confidence adversarial samples are less likely to transfer (Abdullah et al., 2021b). In total, we create 4050 adversarial audio samples.

**Transferability:** Having created the adversarial samples for each of the five ASRs, we transfer them to the remaining four models. Targeted transferability is successful if both the surrogate and the target ASRs output the same target label. However, adversarial samples exhibit asymmetry (Wu et al., 2018; Wu & Zhu, 2020), which is when an adversarial sample generated for model A transfers to model B, but not the other way around. To account for this, we average the number of successfully transferred samples between all five models. This final average is the transferability rate.

## 4 Experimental Analysis

Having described our design choices, we can now run the ablation study to observe the impact of each potential factor. This will involve changing or removing (as needed) each of the potential factors and recording the change in the transferability rate. It is important to note that this does not impact ASR accuracy because of two reasons. First, we retrain the model every single time. Second, the modified architecture resembles one used in current literature. We will point to these papers in each section.

### 4.1 Control Experiment

**Setup:** For the ablation study, we run a control experiment to get baseline transferability rates. We use the same design choices (dataset, model architecture, attack formulation, adversarial audio generation, and transferability) from the previous section. This pipeline resembles the generic ASR pipeline attacked in current adversarial literature (Hannun et al., 2014). We train and attack five instances of the control model and calculate the corresponding transferability rate. The ASRs achieve an average accuracy of 95% and an agreement of 92%.

**Results:** Figure 2(a) shows the transferability rate for our control experiment which is at 42%. Prior work has shown that transferability of optimization attacks is close to *impossible* in real-world ASRs (approximately 0%) (Abdullah et al., 2021b). We show that transferability is still possible for *very* simple ASRs – too simple for the real-world.

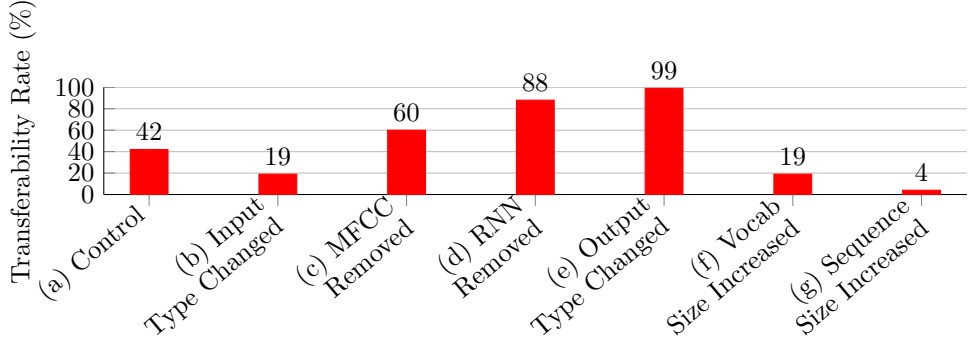

Figure 2: The target transferability rate of optimization attacks for every potential factor. We compare each factor's impact on the transferability rate against the (a) control experiment. There are three key insights. First, image domain samples (b) are harder to transfer than audio ones. Second, removing (c) MFCC, (d) RNN, and (e) sequence output type improves transferability. Third, increasing the (f) vocabulary and (g) sequence sizes reduces transferability.

Having measured the baseline results, we can now study the impact of each of the potential factors (Section 2). We will *only* modify one potential factor at a time while keeping the rest of the control setup unaltered. This will ensure that the change in the potential factor alone affects the transferability rate.

## 4.2 Input Type Changed

**Setup:** Here, we will switch from the audio dataset to an image one. We use the same setup as the one used for the control experiment, except for the training data. Instead of training the ASR on audio samples containing numbers, we train it on images of handwritten numbers from the NIST Special Database (Grother, 1995). This dataset is similar to the MNIST (LeCun, 1998), except that the size of the images is 128x128. To match the one-dimensional shape of audio (vectors of size 16,000), we rasterize the images into single vectors of length 16,384. We did not use the MNIST dataset because its smaller-sized images (28x28) rasterize to a length of only 784, which is substantially shorter than the audio samples. We train five instances of the ASR to a mean accuracy of 93% and agreement of 92%.

**Results:** The results show that the input type does impact the transferability rate, with images being harder to transfer than audio samples. Specifically, the rate falls from 43% for audio (Figure 2(a)), to 19% for images (Figure 2(b)). This fall can be attributed to the contrast in the composition of audio and image samples. Specifically, image samples are composed of feature correlations that become periodic when we convert the images into one-dimensional vectors. Pixels that were next to each other in the original image are far apart in the vector. This can impact the MFCC extracted features and the corresponding robustness of the decision boundary.

## 4.3 MFCC Removed

**Setup:** In this experiment, we will observe the MFCC's effect on transferability by removing it from the pipeline. We use the same setup as the one used for the control experiment, except for the model architecture. We replace the MFCC with trainable layers, effectively converting the ASR into an end-to-end model seen in prior work (Amodei et al., 2016). It is important to note that this end-to-end model has greater complexity due to the inclusion of additional trainable layers. However, the MFCC too acts as a pre-trained, frozen layer since it contains handcrafted mel-filters (Mannell, 1994). In contrast, the new trainable layer

allows learning filters during training. We train five instances of the is ASR to an average accuracy of 95%, and an agreement of 94%.

**Results:** The presence of the MFCC does hinder transferability (Figure 2). We can see this as the rate increases from 42% (Figure 2(a)) to 60% (Figure 2(c)). Further experiments in the Appendix demonstrated that this is because the MFCC is regularizing the feature vector, enables greater robustness to transferability.

## 4.4 RNN Removed

**Setup:** We remove the RNN from the pipeline and observe the change in the transferability rate. We use the same setup as in the control experiment, except we modify the model architecture: we replace the RNN with a convolutional layer to maintain the approximate number of trainable parameters. This converts the pipeline from a sequence-to-sequence mapping type, to a repeated one-to-one mapping type and resembles valid a speech pipeline proposed in other works (Collobert et al., 2016). We train five instances of the ASR, to an average accuracy of 94%, and an agreement of 91%.

**Results:** Removing the RNN from the ASR improves the transferability rate from 42% (Figure 2(a)) to 88% (Figure 2(d)). This demonstrates that the RNN does limit transferability. It is also important to note that we use a vanilla RNN cell in the control, which we replace with a CNN for this experiment. Furthermore, real-world ASRs use more complex RNN cells (e.g., GRUs and LSTMs) that include additional weights. Since model complexity hinders transferability (Wu et al., 2018; Demontis et al., 2019), we expect that the additional parameters of the GRU and LSTM cells will decrease transferability even further, enabling even greater robustness in real-world models.

## 4.5 Output Type Changed

**Setup:** To study the impact of the sequence output type, we replace it with a single-label one. Instead of outputting a sequence of characters per frame, the pipeline will output a single classification label (ONE to NINE) for the entire audio. We use the same setup as the one used for the control experiment, except we modify the final layer of the model architecture. The resulting ASR will produce a single output for an entire input, effectively converting the pipeline from a sequence-to-sequence mapping to a one-to-one type. While this setup may seem unusual, speech pipelines have used this approach in past works (Chen et al., 2014; Higgins & Wohlford, 1985; Rose & Paul, 1990).

**Results:** The transferability increased from around 42% (Figure 2(a)) to almost 99% (Figure 2(e)). This rate is almost identical to the 100% we observed in the image domain (Supplementary Materials Section A.3) and suggests that task type plays a very significant role in the transferability rate. A model that uses a sequence labeling task, instead of a single label one, will be more robust to transferability. Additionally, sequence labeling is necessary for ASRs due to the variability of speech. Therefore, training an ASR as in the current experiment is not possible for anything but trivial systems.

## 4.6 Vocabulary Size Increased

**Setup:** The output complexity can be measured using two variables, one of which is vocabulary size (Section 2.3.4). In this experiment, we observe the impact of vocabulary size by increasing its value. We use the same control setup, except that we increase the total number of output labels from 9 to 30. We do so by using the full Google Speech Commands dataset. We train five instances of the ASR, to an average accuracy of 92%, and agreement of 87%. This reduction in agreement from 92% (control experiment) to 87% is normal. By increasing the number of output labels, we increase the likelihood of even the benign samples being labeled incorrectly.

**Results:** We compare the control (trained on 9 output labels) with this modified setup (trained on 30 output labels). We can observe that the transferability rate fell by 24 points, from 42% (Figure 2(a)) to 19% (Figure 2(f)). Certainly, part of this drop can be attributed

to the fall in agreement, from the control (92%) to the current setup (87%). However, a five percent drop in the agreement can not alone result in a large 24 percent drop in transferability. Therefore, a we believe that a substantial degree of the drop is due to the increased vocabulary size. Furthermore, real-world data sets contain hundreds of thousands of unique words (Panayotov et al., 2015), far more than the 30 we tested in this experiment. Therefore, we can expect a further decrease in transferability in more realistic training data sets.

## 4.7 Sequence Size Increased

**Setup:** In addition to vocabulary size, the sequence size also contributes towards output complexity. Similar to the previous experiment, we will study the impact on sequence size by increasing its value. We use the same setup as we had in the control experiment, except that we modify the original numbers-only training dataset. We generate a new dataset containing sequences of four numbers by concatenating audio samples (and their corresponding labels). For example, we concatenate four audio files that contain the words ONE, NINE, ONE and NINE, resulting in a single audio file containing the sequence ONE NINE ONE NINE. We train five instances of the ASR to an average accuracy of 95%, and agreement of 86%.

**Results**: Increasing the size of the sequence by just a single word can reduce the transferability rates from 42% (Figure 2(a)) to a mere 4% (Figure 2(g)). Real-world ASRs process much longer sequences than just size four. The average length of a sequence in the English language is around 15 words (Cutts, 2020). Therefore, due to the increased sequence size, we can expect an even further decrease in the transferability rate in real-world ASRs.

## 5 Discussion and Takeaways

In the previous section, we explore how the six factors impact the transferability of optimization attacks in ASRs. In this section, we will discuss the takeaways from our findings.

**An ablation study to explore factors impacting transferability *must* be performed on a simple, but realistic ASR pipeline.** Real-world ASRs are large and complex, which itself limits the transferability of optimization attacks to an abysmal 0%. Performing an ablation study on such models will yield no results since removing or changing any component will not change the 0% transferability rate. This will hide the impact of other unknown factors that might be playing a role. As a result, we run an ablation study on a simple ASR pipeline and uncover six previously unknown factors that limit transferability.

**Existing optimization attacks are unlikely to provide targeted transferability against real-world ASRs.** In this paper, we uncover a number of factors that limit the transferability of optimization attacks in ASRs. Interestingly, these same factors are known to improve accuracy, and therefore, will likely be found in most real-world ASRs (Hannun et al., 2014). As a result, real-world ASRs will likely remain robust to optimization attacks, motivating the need to our attention to other attack types.

**The factors preventing transferability are also required for the correct functioning of the ASR.** Output type, large vocabulary, and sequence sizes are crucial for training accurate general-purpose ASRs. This is because ASRs need to account for varying audio length (output type) and a variety of words (vocabulary size) and phrases (sequence sizes). It might not be possible to train any real-world ASR without these three components. Fortunately, these same components also prevent transferability. Therefore, real-world ASRs will be robust to the targeted transferability of existing optimization attacks.

**Seeking better attacks:** Since optimization attacks do not provide targeted transferability in ASRs, the community should focus on the attack types that can. Signal processing attacks (Abdullah et al., 2021a; 2019) is a family of attacks unique to the audio domain. These attacks exploit the feature extraction layer of the ASR pipeline (Figure 1(b)) and therefore, provide key advantages. These attacks not only provide targeted transferability, but also require fewer queries, are model agnostic, require only black-box knowledge, and take mere seconds to execute (Abdullah et al., 2021b). However, clean, targeted signal processing attacks (i.e., adversarial audio that sounds clean to humans and are transcribed

as the targeted text) do not yet exist. Therefore, developing such attacks (to replace optimization ones) is a potential direction for future research.

**Building speaker recognition models that are robust to targeted transferability of optimization attacks:** Speaker recognition is one application in the audio domain that has a very similar pipeline to that of ASRs. However, unlike ASRs, speaker recognition models (specifically text-independent ones) are not robust to targeted transferability of optimization attacks (Chen et al., 2019). Text-independent models check for speaker identity, without verifying the text in the audio sample. As a result, these models do not employ sequence labeling, vocabulary, and sequence sizes, the absence of which enables transferability. As a result, it is possible to build robust speaker recognition models by simply including text verification as part of the pipeline. This will involve using an ASR for text verification, which can then be followed by speaker identification. Since target transferability will fail at the ASR level, the speaker identification will not be triggered. This will ensure the robustness of the overall speaker recognition pipeline.

# 6 Related Work

ASRs are vulnerable to adversarial samples that can force them to output malicious labels. In the audio domain, there are three types of attacks to generate such adversarial samples: signal processing attacks (Abdullah et al., 2021a; 2019), gradient-free attacks (Taori et al., 2018; Alzantot et al., 2017; Chen et al., 2020), and optimization attacks (Carlini & Wagner, 2018; Cissé et al., 2017; Kreuk et al., 2018a; Qin et al., 2019; Schönherr et al., 2019; Abdoli et al., 2019; Yuan et al., 2018; Yakura & Sakuma, 2018; Alzantot et al., 2017).

**Signal Processing Attacks:** do exhibit targeted transferability because these exploit the feature extraction stage of the ASR pipeline. They produce adversarial samples whose feature vectors are similar to the ones produced for the benign sample. As a consequence, the model is unable to ascertain whether the feature vector came from benign or adversarial audio. However, there do not yet exist signal processing attacks that can produce clean audio that is also transcribed as the adversary chosen text. So far, existing signal processing attacks can only generate targeted noisy audio (i.e., the attack audio sounds like noise to the human ear, but is transcribed as the targeted text by the model) or untargeted clean audio (i.e., the attack audio sounds clean to the human ear, but is transcribed as garbage text by the model).

**Gradient-Free Attacks:** craft adversarial samples by repeatedly querying the target model. Theoretically, this can enable attackers to exploit models in black-box settings. However, these attacks have not had any success against real-world ASRs. Additionally, these attacks have not demonstrated any transferability, even against the simple ASRs.

**Optimization Attacks:** use the model gradients to craft adversarial samples. These attacks have been incredibly successful in the image domain (Szegedy et al., 2013; Goodfellow et al., 2014; 2016; Papernot et al., 2017; 2016). One of the reasons for this is the ability of their samples to exhibit transferability even for real-world black-box systems (Liu et al., 2016). Most of the work in the space of transferable optimization attack samples has therefore been focused on image recognition models. There are several factors that have been shown to affect transferability, which we list in Section 2.1. In contrast, our work specifically focuses on the audio domain. We are the first to study the factors that make the targeted transferability of optimization attacks difficult against ASRs.

# 7 Conclusions

In this work, we investigated why ASRs robust to target transferability of optimization attacks. We conducted an exhaustive ablation study and uncovered previously unknown factors that limit transferability. The ability of these factors to limit transferability exposes a serious limitation in optimization attacks against ASRs. As a result, attack types that have demonstrated transferability in real-world settings, like signal processing attacks, deserve more attention.

## 8 Acknowledgments

This work is partly supported by the National Science Foundation under CNS-1933208. Any opinions, findings, and conclusions or recommendations expressed in this material are those of the authors and do not necessarily reflect the views of the National Science Foundation. Additionally, we would like to thank Patrick Emami for providing valuable comments and revisions on an early draft of this work.

## 9 Code of Ethics

These authors have reviewed and adhered to the ICLR Code of Ethics.

## 10 Reproducibility Statement

To train the ASR, the readers can refer to the pipeline found in the Supplementary materials, Figure 3. This figure describes the exact ASR architecture we used in the experiments. The information about the training data, ASR pipeline, attack formulation, and attack hyper parameter details exist in Section 3.

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

## A  Supplementary Materials

### A.1  RNN as a Potential Factor

A typical ASR neural network is made up of the following four stacked components:`CNN->RNN->FC->LOGITS`, as seen in Figure 1(iii). It is not necessary to exhaustively study the impact on the transferability of each of these components. This is because transferability in image models is far easier (Supplementary Materials A.1) than in ASRs Abdullah et al. (2021b). This suggests that there are component(s) unique to ASRs (i.e., components

non-existent in image models) that might be hindering transferability. As a consequence, we focus our study on the symmetric difference of the two model architectures (i.e., present in ASRs but not in image models). The only component that fits this criterion is the RNN, as shown in Figure 1(c).

## A.2 REAL WORLD ASR FOR ABLATION STUDY

In the ideal case, we would want to run the ablation study on real-world ASRs, since we are trying to make conclusions about such ASRs. However, such a study will provide inconclusive results about the potential facotors. A real-world ASR has high model complexity, with tens of millions of training parameters. Similarly, it is trained on complex data sets (i.e., containing a large vocabulary size of hundreds of unique words and long sequence sizes). We already know that large model and output complexity limit transferability to 0% Abdullah et al. (2021b). As a consequence, if we were to run an ablation study on such an ASR, the model and output complexity alone will force the transferability to remain at 0%. For example, while removing the RNN increases transferability (Figure 4), this step will have no impact on transferability in the case of a real-world ASR.

To validate this hypothesis, we conducted a simple experiment. We trained multiple real-world ASRs (on identical setups) that did not contain an RNN using the wav2letter architecture Collobert et al. (2016). We used the LibriSpeech data set for training, which contains 1000 hours of audio, has a vocabulary size of 900,000 unique, and up to a sequence size of 20. We followed the same methodological steps outlined in Section 3 to produce the adversarial samples. Of these, *none* of them transferred from the surrogate to the target models we had trained.

As a consequence, we had no choice but to run the study this phenomenon on a simpler ASR using a less complex training data set. This limits the impact of the known factors and helps isolate the impact of the unknown ones.

## A.3 TARGETED TRANSFERABILITY FOR IMAGE MODELS

We know that targeted transferability rates for audio models are abysmally low Abdullah et al. (2021b), even when the models are trained on identical setups. We conduct the following experiment to study whether this is true for image recognition models:

### A.3.1 SETUP

We use the same general setup we had used in the control experiment (Section 4.1), except change the architecture, the training data, and the attack. We train on the MNIST datasetLeCun (1998) on the following architecture: `CNN->MAXPOOL->CNN->MAXPOOL->CNN->FC->LOGITS`. The model contains 250,000 trainable parameters, similar to the control. We train the models for 12 epochs, with 1024 batch size an accuracy of 94% and an agreement of 89%. We use the basic iterative attack Goodfellow et al. (2014) clipping perturbations at values of 0.1 and 0.05.

### A.3.2 RESULT

We observe a transferability of exactly 100% for both clip values, for all iterations. In stark contrast, audio transferability is close to 43% (Section 4.1). This is despite the fact that the attack we use against the image models uses clipping. In contrast, we perform no such clipping in the audio attack (Section 3.4). To add to that, the accuracy and agreement of the image model (94% and 89%) is lower than the control (95% and 92%). This implies that there are additional previously unknown factors that are preventing transferability in the audio domain.

## A.4 MFCC AND TRANSFERABILITY FOR ASRS

transferability rate remains constant for the control, when the MFCC is present (Figure 5(a)). However, when the MFCC is removed, the transferability rate drops from 65% (at

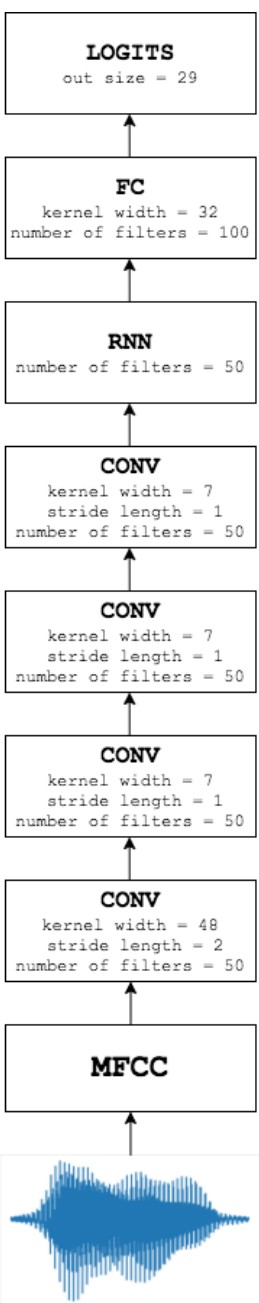

Figure 3: The details of the pipeline we used in the control experiment. The pipelines for the potential factors we study in the paper are all based on the one shown above.

50 iterations) to 57% (at 500 iterations), shown in Figure 5(b). This is because increasing iterations produce "over-fitted" adversarial samples. This suggests that the MFCC is regularizing the feature vector, whereby helping the model learn smoother decision boundaries robust to target transferability.

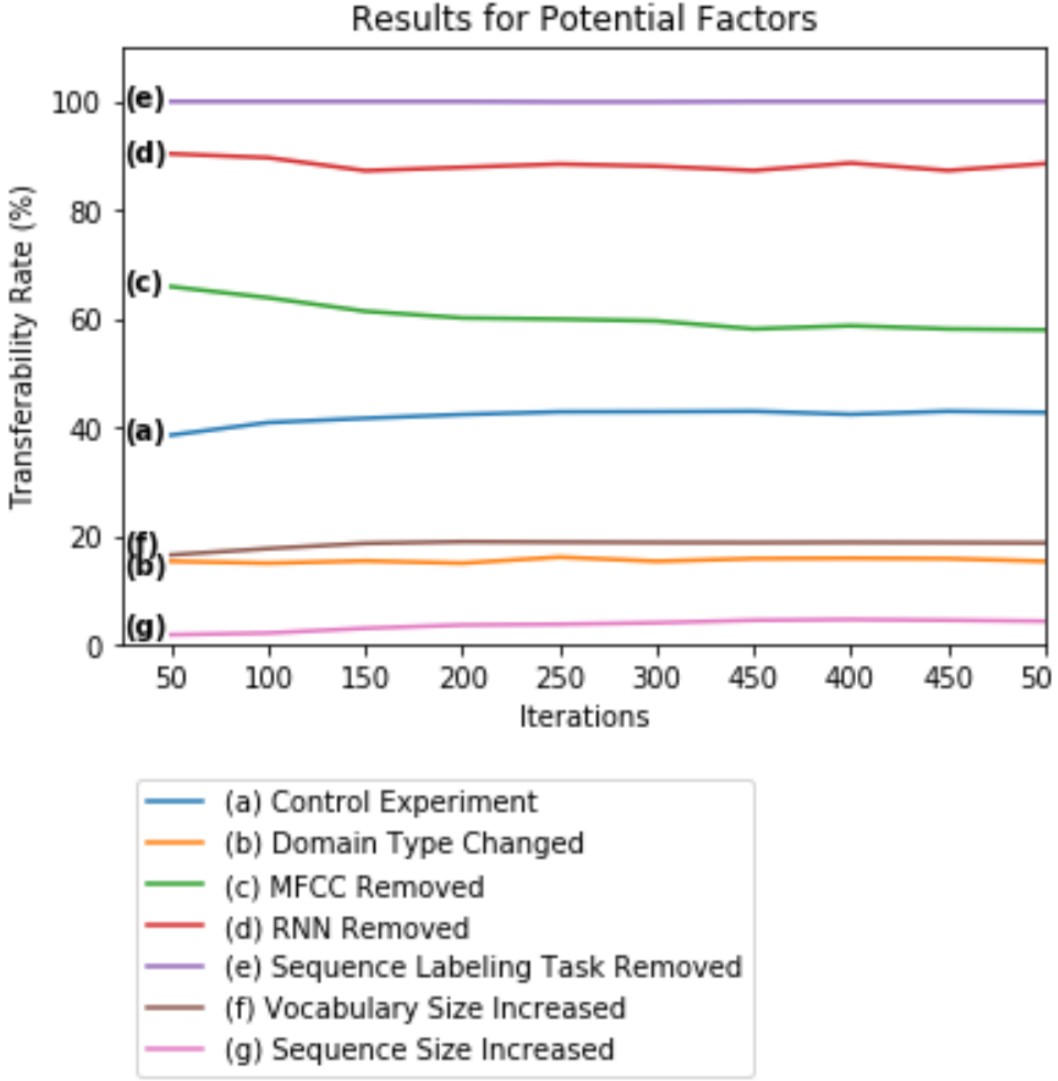

Figure 4: We can observe the change in the transferability rate with respect to the number of iterations. Generally, we see that the number of iterations does not have a significant impact on the transferability rate, except when the MFCC is removed.

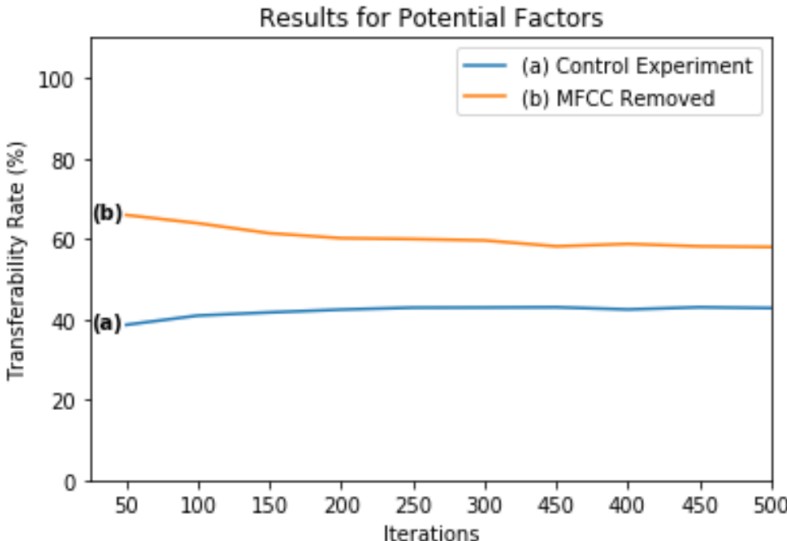

Figure 5: The plot shows the relationship between transferability and iterations for two setups: the control experiment and when the MFCC is removed. For the control experiment the transferability rate is consistent across all iterations (a). However, when the MFCC is removed, the rate falls as the iterations increase (b). This suggests that the MFCC is regularizing the feature vector helping the model learn smooth and robust decision boundaries.

