# OpenReview forum: "Demystifying Limited Adversarial Transferability in Automatic Speech Recognition Systems"
_ICLR.cc/2022/Conference — ICLR 2022 Poster_

### Official Review · Reviewer_qpFd · 2021-11-02

**Correctness:** 3
**Technical Novelty And Significance:** 2
**Empirical Novelty And Significance:** 2
**Recommendation:** 5
**Confidence:** 4

**Main Review:**

The authors design a simplified ASR system in order to show which components of
such a system are affecting the targeted transferability issue. First, a
simplified ASR system is required, as it has yet to be shown that complex
real-world ASR systems have 0% targeted transferability.

The writing is clear, and the design is simple. The results provide some evidence
as to which components contribute to targeted transferability.

However, some of the choices in the modeling and explanations are a little weak in
some areas:

1. The model is not fully specified within the text, only mentioned in a reference.
The model should be at least briefly summarized. In the least explain how the model
goes from frame level output labels to a label (CTC? RNN-T?).

2. The authors discuss vocabulary size, but is this even handled within their model?
The model to me appears to only have 9 labels which are synonymous with vocabulary size.
Also, since there are no details of the model within the paper, how is the alignment issue handled?

3. What about attacks on the mfcc themselves?

4. Why wasn't there an ablation study on more complex RNN cells (LSTM, GRU). Should have been straightforward
to run this test.

5. Why is there no ablation study on the effects of dropout?

Smaller points:

a. In the description of input types, what is the significance of having topographical input with respect
to adversarial attacks.

b. In 2.3.4, " the victim ASR needs to assign the attacker chosen character to each frame of the adversarial audio",
but this is highly dependent on the ASR model. For example in CTC or RNN-T, what excludes the possibility of
modifying only a handful of frames in order to affect the output.

c. In 2.3.5 the frame level output for your simplified ASR model is indeed fixed. The output sequence is also
fixed (albeit large).

d. No reference for Deepspeech.


**Summary Of The Paper:**

It is known that targeted transferability is ineffective against typical ASR
systems, but it is not fully understand why.  In this work the authors show the
impact of targeted transferability against a simplified ASR system.
Specifically, they perform an ablation study to assess which components of an
ASR may be contributing to the non targeted transferability issue.  Their
results provide insights into ASR case, which may help against other
transferable attack types, and also help in other domains.


**Summary Of The Review:**

I believe the authors provide a useful beginning of understanding why ASR systems may be
not susceptible to targeted transferability. However, I think a little further work
could have been done to improve the results as well as the explanations.

---

> ### Author Response · Authors · 2021-11-22
> **Point by point rebuttal.**
>
>
> We thank the reviewer for their comments.
> 1. We placed the full architecture in the Appendix (Figure 3) due to space constraints. We will also be happy to provide a brief summary of the architecture and its alignment algorithm (CTC) in the camera-ready version of the paper.
>
> 2. We train two different models with vocabulary sizes of 9 and 30 (Section 4.6). The goal of this experiment was to study the relationship between vocabulary size and transferability rates. The results showed that the transferability rates fall as vocabulary increases. Regarding alignment issues, we used the DeepSpeech architecture. This is the most commonly attacked ASR architecture in current literature [6] and uses CTC to deal with alignment problems.
>
>
> 3. We assume that the reviewer is referring to the signal processing attacks [1,2] as the attacks on the MFCC. Signal processing attacks have very high transferability. This is because they exploit the signal processing-based feature extraction layer of the ASR. Our work, instead, focuses primarily on optimization attacks and why they fail to transfer.
>
>
> 4. Due to additional neurons and gates, LSTM and GRU cells have higher complexity than vanilla RNN cells. And we know from prior work, higher complexity reduces transferability [3,4,5]. In our experiments, we wanted to answer whether the presence of even a simple recurrent layer with low complexity (like the vanilla RNN) would impact transferability. Our results show that having any recurrent layer, even a simple one, limits transferability. If we were to repeat the experiments with the more complex GRU or LSTM, the increased complexity would limit transferability even further (more so than the vanilla RNN). We provide these details in Section 4.4 and will be happy to clarify them in the camera-ready version of the paper.
>
>
> 5. The effects of the dropout on transferability are well known in the community [3]. In fact, some of our earlier experiments (not included in the paper) cemented this idea. As a consequence, we did not see the need to include it in the paper.
>
>
> Smaller Points:
> We thank the reviewer for the smaller points. The reviewer is correct. Our experiments used DeepSpeech (a CTC-based ASR), which is the most commonly attacked ASR architecture in current literature [6]. We appreciate this comment and acknowledge our mistake. We will be happy to include a reference to DeepSpeech.
>
> [1] Hear "No Evil", See "Kenansville": Efficient and Transferable Black-Box Attacks on Speech Recognition and Voice Identification Systems. 42nd IEEE Symposium on Security and Privacy, 2021. Hadi Abdullah, Muhammad Sajidur Rahman, Washington Garcia, Logan Blue, Kevin Warren, Anurag Swarnim Yadav, Tom Shrimpton, Patrick Traynor.
>
>
> [2] Practical Hidden Voice Attacks against Speech and Speaker Recognition Systems. Network and Distributed System Security Symposium (NDSS), 2019.
> 2nd place at CSAW'19 Applied Research Competition. Hadi Abdullah, Washington Garcia, Christian Peeters, Patrick Traynor, Kevin Butler, and Joseph Wilson.
>
>
> [3] Ambra Demontis, Marco Melis, Maura Pintor, Matthew Jagielski, Battista Biggio, Alina Oprea, Cristina Nita-Rotaru, and Fabio Roli. Why Do Adversarial Attacks Transfer? Explaining Transferability of Evasion and Poisoning Attacks. In 28th {USENIX} Security Symposium ({USENIX} Security 19), pp. 321–338, 2019.
>
> [4] Lei Wu, Zhanxing Zhu, Cheng Tai, et al. Understanding and Enhancing the Transferability of Adversarial Examples. arXiv preprint arXiv:1802.09707, 2018.
>
> [5] Lei Wu and Zhanxing Zhu. Towards understanding and improving the transferability of adversarial examples in deep neural networks. In Asian Conference on Machine Learning, pp. 837–850. PMLR, 2020.
>
> [6] The Faults in our ASRs: An Overview of Attacks against Automatic Speech Recognition and Speaker Identification Systems. 42nd IEEE Symposium on Security and Privacy, 2021. Hadi Abdullah, Kevin Warren, Vincent Bindschaedler, Nicolas Papernot, and Patrick Traynor.

---

> > ### Comment · Reviewer_qpFd · 2021-11-29
> > **Good clarifications.**
> >
> > Thank you for the clarifications on the points I raised.
> >
> > 1. I failed to see the architecture the first time, but see it now.
> > 2. The comparison in 4.6 makes sense now with respect to the vocabulary size. And thank you for clarifying the alignment question.
> > 3. Thank you for references.
> > 4. This explanation makes sense, as even simple RNN's impact transferability.
> > 5. Thank you for references.

---

### Official Review · Reviewer_TNL7 · 2021-11-03

**Correctness:** 3
**Technical Novelty And Significance:** 3
**Empirical Novelty And Significance:** 3
**Recommendation:** 5
**Confidence:** 3

**Main Review:**

# Strengths
- A timely study on an important problem has been highlighted recently.
- Ablation evaluation methodologies are systematic.
- The finding on robustness improvement and transferability is interesting.

# Weaknesses
- Lack of in-depth root causes analysis on the findings. The paper provides some discussion in each ablation study. However, the current discussion does not go beyond describing the observations of the experimental results. For measurement papers, the readers are more interested (and will be more appreciated) if an in-depth analysis on the fundamental reasons why such feature affects the transferability. This is not an easy task, where additional small-scale experiments may also be required as the authors generate hypotheses.
- The current discussion of the findings is limited to the ASR domain, however, as ASR and image tasks were discovered to have distinctive attack transferability properties, the author may consider drawing connections with the image space. For example, will the similar features also make transferability more difficult in the image tasks? Such analysis also helps to understand the unique challenges in the ASR and perhaps improve the robustness of image task models.

**Summary Of The Paper:**

The paper conducts a systematic study on the phenomenon that attacks targeting ASR systems often have low transferability. To do that, the authors take a representative ASR pipeline and perform ablation studies by modifying or removing the components that may have an effect on the attack transferability. Results show that many existing designs for improving the robustness of ASR can also prevent transfer attacks. Based on the findings, the authors also discuss the takeaways and future directions for the ASR.

**Summary Of The Review:**

The paper studies an important problem and presents many interesting findings. However, it still lacks root cause analysis on the findings to indeed “demystify” the low attack transferability in ASR.

---

> ### Author Response · Authors · 2021-11-22
> **Point by point rebuttal.**
>
> Lack of Depth:
> We thank the reviewer for this comment. We understand that the readers will be interested in learning the why, not just the how. Therefore, we did provide explanations as to why certain components limit transferability. For example, we mention that MFCCs reduce transferability as they regularize the input [1,2,3,4]. Similarly, we point out that large vocabulary and sequence sizes limit transferability because they increase the model's output complexity. This is in congruence with prior research, which shows that models with high output complexity have lower transferability [1,3,4]. Similarly, we describe in the text how having a sequence output type decreases transferability since it is easier for the model to make mistakes. This is because the model needs to assign the exact sequence label. Making a single mistake of assigning one wrong label in the entire sequence can change the semantic meaning completely (e.g., changing the 'h' to a 'c' in 'hat' will change the word to 'cat'). We will be happy to clarify this in the camera-ready version of the paper.
>
> Connection to Images:
> We really appreciate this comment and do see distinct connections to the image space, specifically video captioning. Similar to ASRs, these models have been shown to be more robust to transferability than generic image recognition models [5]. While no work has focused on explaining this lack of transferability in video captioning models, our results from ASRs do point to a potential reason. Much like ASRs, video captioning models also use RNNs. Our work shows that RNNs reduce the transferability rates in ASRs, and will therefore also reduce transferability in video captioning models as well.
>
> Similarly, we show that large vocabulary sizes limit transferability because they increase the model's output complexity. This has a direct connection with the image domain. Models trained on the MNIST dataset (10 labels) have much higher transferability than ones trained on the Imagenet dataset (1000 labels). We will be happy to provide these details in the camera-ready version of the paper.
>
> [1] Ambra Demontis, Marco Melis, Maura Pintor, Matthew Jagielski, Battista Biggio, Alina Oprea, Cristina Nita-Rotaru, and Fabio Roli. Why Do Adversarial Attacks Transfer? Explaining Transferability of Evasion and Poisoning Attacks. In 28th {USENIX} Security Symposium ({USENIX} Security 19), pp. 321–338, 2019.
>
> [2] Wen Zhou, Xin Hou, Yongjun Chen, Mengyun Tang, Xiangqi Huang, Xiang Gan, and Yong Yang. Transferable adversarial perturbations. In Proceedings of the European Conference on Computer Vision (ECCV), pp. 452–467, 2018.
>
> [3] Lei Wu, Zhanxing Zhu, Cheng Tai, et al. Understanding and Enhancing the Transferability of Adversarial Examples. arXiv preprint arXiv:1802.09707, 2018.
>
> [4] Lei Wu and Zhanxing Zhu. Towards understanding and improving the transferability of adversarial examples in deep neural networks. In Asian Conference on Machine Learning, pp. 837–850. PMLR, 2020.
>
> [5] SK Adari, W Garcia, K Butler. Adversarial Video Captioning. 49th Annual IEEE/IFIP International Conference on Dependable Systems and Networks Workshops (DSN-W), 2019.

---

### Official Review · Reviewer_9PZL · 2021-11-03

**Correctness:** 2
**Technical Novelty And Significance:** 2
**Empirical Novelty And Significance:** 3
**Recommendation:** 5
**Confidence:** 4

**Main Review:**

Strengths:

+ The paper is well-written and easy to follow.

+ The paper gives some takeaways about the limited adversarial transferability in ASR systems.

+ The paper does lots of work for the system implementation and testing.

+ The paper gives many details in implementation and evaluation.

Weaknesses & comments:
- I basically like the ideas and findings presented in this paper. However, I think this paper focuses on the measurement of ASR systems and may be more suitable for a security conference.

- The method of how the authors come up with the factors is not explained. This is the main limitation of the paper.

- The definition of a “factor” is not clear and is not clearly defined in this paper. For example, RNN is a factor. Also, Output Type is a factor, but specifically, Sequential Output is a factor/reason? Same with Input Type, and Output Complexity. Is Vocabulary a factor, or is Complexity a factor? Isn’t the total number of Output Labels factor the same as Output Type factor? So please define what is a factor. Please also see my comment on EFA.

- As a consequence of the above point, don’t you think “model complexity” is also a factor? Maybe even experiment with transferability Vs. increasing model complexity?

- How does “Number of models” simultaneously optimized in Equation 1 affect transferability? Do you think this could improve transferability? (e.g. optimize jointly on 3 models, test on 4th model).

- The ASR pipeline in Section 2.2 should be generalized. For example, call the RNN as “ASR model”, MFCC as “Signal Processing Algorithm” and so on. Also, how did the authors give this particular pipeline as an example? Is it something that they have observed in existing literature?

- I would like to see an EFA analysis on these factors. What are the groupings of these factors?

- The second factor, MFCC, should be generalized. I am not an expert in this field, but I suppose there are more algorithms than just MFCC that do this operation.

- I don’t understand the RNN factor, which is not clear. Is the RNN itself a factor, or any sequence model, such as RNN, LSTM … a factor?

- The current evaluation is not extensive. The paper can benefit from giving a deeper evaluation of these identified factors. For example, what are the detailed impacts of each known factor in the control experiment? What will happen if you change two factors at the same time, e.g., remove MFCC but use a complex RNN? And what is the impact of using RNN with different complexity?


**Summary Of The Paper:**

This paper presents an evaluation and explanation of the limited adversarial transferability in the ASR system. It first lists 11 known factors, e.g., smoothness of gradients, and then proposes four potential factors that limit the transferability. Then it examines the transferability in the DeepSpeech system and found its model is too complex for adversarial transferability. Hence, the author implements 5 ASR systems based on a simple model trained from the Google Speech Commands dataset. It generates 500 adversarial audio samples for each of the five ASRs and tries to transfer samples to others. The evaluation examines the target transferability rate of optimization attacks by one factor at a time. Then it gives some insights into the impact of each of the potential factors.

**Summary Of The Review:**

Although this paper looks simple, it has actually identified an interesting, novel problem. I think the authors motivated this paper from a recent SOK paper published at the IEEE S&P'21 conference that has discussed related issues. I appreciate that the authors have done research on such an important topic. I would suggest accepting this paper as a short paper or a poster if possible.

---

> ### Author Response · Authors · 2021-11-22
> **Point by point rebuttal.**
>
> We thank the reviewer for their detailed comments.
>
> “The method of how the authors come up with the factors is not explained…”
> We used the following methodology:
>  Since image models and ASRs had widely different transferability rates, we hypothesized that ASRs have additional components (which do not exist in image models) that might limit transferability. We then analyze the ASR pipeline, identify 6 such components, and test our hypothesis with the ablation study.
>
> “The definition of a “factor” is not clear…”
> We understand the reviewer’s point regarding the choice of the word “factor”. We will be happy to use a different word, such as attribute or component, in the final version of the paper. Additionally, the reviewer is correct to point out that output complexity is a factor. However, since ASRs output multiple labels for a given input, ASR output complexity can be broken into two categories: vocabulary size and sequence size. We treat each of them as separate factors to understand their individual impact on the transferability rates.
>
> “As a consequence of the above point, don’t you think “model complexity” is also a factor?....”
> We agree with the reviewer that model complexity is a factor. Infact, we cite it as one of the well-known factors from existing literature that limits transferability (Section 2.1). We observed its impact during our initial experiments against a complex real-world ASR (Appendix A2). Due to the sheer complexity of the model, the transferability rate was 0%. Infact, complexity is such a strong factor that even during the ablation study (when we removed components) of this real-world ASR, the transferability did not increase from 0%. Therefore, we were compelled to use a simpler, less complex model for the remaining experiments of the paper.
>
> “How does “Number of models” simultaneously optimized in Equation 1…”
> Prior work in the image space has shown that jointly optimizing models does improve transferability. However, this experiment (though interesting) is not within the scope of this work. Our primary goal is to identify the factors limiting transferability in ASRs.
>
> “The ASR pipeline in Section 2.2 should be generalized…”
> We appreciate this comment and will be happy to make the changes in the camera-ready version of the paper. Additionally, we used this specific pipeline (namely DeepSpeech) because it is the most commonly attacked one in existing literature [1].
>
> “The second factor, MFCC, should be generalized…”
> We appreciate this comment. We chose the MFCC based ASR is it most commonly studied one in current attack literature. Other possible feature extraction methods include the FFT, bark scaled features, and mel scaled features. However, these are nearly identical to the MFCC, with only minor variations. Therefore, our overall result (that feature extraction reduces transferability) will still hold, even if we use different feature extraction methods. We will be happy to clarify this in the text.
>
> “I don’t understand the RNN factor”
> We acknowledge that our description is unclear. Yes, any sequence model or recurrent node (LSTM, GRU, vanilla RNN) is a factor. This is because in our experiments, we simply removed the memory state from the RNN, which effectively removed the sequence/recurrence relation. Our experiments show that it does not matter what specific recurrent node you use (vanilla rnn or a complex LSTM/GRU), as long as a memory state is present, transferability will remain low.
>
> “The current evaluation is not extensive…”
> We thank the reviewer for this comment. The effects of the known factors are already understood in existing literature (Section 2). Therefore, we are not sure what additional insight we will gain from repeating those experiments. This includes model complexity, which is known to decrease transferability. Similarly, removing multiple factors will not reveal any additional information. The 5 factors we identify all reduce transferability. If we remove subsets of these factors, the reduction in transferability will be even starker.
>
> [1] The Faults in our ASRs: An Overview of Attacks against Automatic Speech Recognition and Speaker Identification Systems 42nd IEEE Symposium on Security and Privacy, 2021. Hadi Abdullah, Kevin Warren, Vincent Bindschaedler, Nicolas Papernot, and Patrick Traynor.

---

### Official Review · Reviewer_Vmod · 2021-11-04

**Correctness:** 3
**Technical Novelty And Significance:** 3
**Empirical Novelty And Significance:** 3
**Recommendation:** 8
**Confidence:** 3

**Main Review:**

This paper studies an important problem and provides valuable insights into the mysterious fact of the very low target transferability between ASR models. The control experiments are carefully executed and provide clear takeaway insights. The paper is also well-written and nicely structured. Below are my comments that can further improve this paper:
1. The experiments on the input type do not seem very interesting to me. Images are inherently 2-dimensional. Converting them to 1-dimensional naively can destroy the locality. There are more advanced ways to convert and an image to 1D, such as using a Hilbert curve.
2. The paper only uses the vanilla attack. There have been quite a few papers in the adversarial machine learning literature on improving transferability (e.g., Expectation over Transformation, attacking an ensemble, etc).
3. This paper only uses a single model, DeepSpeech, which is a character-based model. A word-based model may have different behaviors.


**Summary Of The Paper:**

This paper studies why optimization-based adversarial attacks have close to zero target transferability in attack real-world ASR pipelines.
The paper identifies 6 previously unknown factors that impact target transferability, including the input type, Mel Frequency Cepstral Coefficient (MFCC), the Recurrent Neural Network (RNN), output type, and the vocabulary and sequence sizes.


**Summary Of The Review:**

A good paper that studies an important problem with carefully executed control experiments.

---

> ### Author Response · Authors · 2021-11-22
> **Answer to the reviewers questions.**
>
> 1. We appreciate the reviewer's comment regarding the input type. However, past reviewers had expressed interest in such an experiment and that is why included it in the paper.
>
> 2. We agree with the reviewer. There have been a number of non-vanilla attacks in the image space that attempt to improve transferability. However, we observe that baseline transferability in ASRs is much lower than in image-based models (Appendix A.3 and [1]). Similarly, non-vanilla attacks (that attempt to improve transferability) will have lower transferability rates in ASRs compared to their image counterparts.
>
> 3. We do explore the use of word-based models in Section 4.7. Instead of assigning characters to each frame, the model assigns words to audio samples. We see that the transferability rate dramatically improves from 42% to 99%. Additionally, our use of DeepSpeech was primarily motivated by the fact that this is the most widely attacked speech model in current literature [1].
>
> [1] The Faults in our ASRs: An Overview of Attacks against Automatic Speech Recognition and Speaker Identification Systems. 42nd IEEE Symposium on Security and Privacy, 2021. Hadi Abdullah, Kevin Warren, Vincent Bindschaedler, Nicolas Papernot, and Patrick Traynor.

---

### Decision · Program_Chairs · 2022-01-20

**Decision:**

Accept (Poster)

**Comment:**

This paper explores why adversarial examples do not transfer well in
adversarial examples on automatic speech recognition systems. The authors
propose a number of potential causes that are then quickly evaluated in
turn.

This could be an excellent paper, but in its current form, it is borderline.
The main problem with the paper is that it proposes a number of causes for the
limited transferability, and then evaluates each of them with one quick
experiment and just a paragraph of text. In particular, none of the results
actually convince me that the claim is definitely correct, and many of the
experimental setups are confusing or would have other explanations other than
the one variable that is aiming to be controlled for.

That said, even with these weaknesses, this paper raises interesting and new
questions with an approach I have not seen previosuly. So while I don't believe
the paper has done much to actually demystify transferability, it does take
steps towards performing scientific experiments to understand why it is so
hard. And these experiments, while not perfect, can serve as the basis for
future work to extend and understand which factors are most important.